# An Indoor Gardening Planting Table Game Design to Improve the Cognitive Performance of the Elderly with Mild and Moderate Dementia

**DOI:** 10.3390/ijerph17051483

**Published:** 2020-02-25

**Authors:** Winger Sei-Wo Tseng, Yung-Chuan Ma, Wing-Kwong Wong, Yi-Te Yeh, Wei-I Wang, Shih-Hung Cheng

**Affiliations:** 1Graduate School of Industrial Design, National Yunlin University of Science and Technology, Douliou, Yunlin 640, Taiwan; mayc@yuntech.edu.tw (Y.-C.M.); Jeff811019@gmail.com (Y.-T.Y.); ny90777111@gmail.com (W.-I.W.); 2Department of Electronic Engineering, National Yunlin University of Science and Technology, Douliou, Yunlin 640, Taiwan; wongwk@yuntech.edu.tw; 3Industrial Design Department, National United University, Miaoli 36003, Taiwan; achille@nuu.edu.tw

**Keywords:** dementia elderly, attention improvement, horticulture therapy, contextual inquiry, indoor gardening table game for dementia elderly

## Abstract

The purpose of this study is to improve the overall cognitive function of patients with dementia in Yunlin County, Taiwan, by designing an indoor gardening flower combination game suitable for home and maintenance institutions. This paper uses qualitative research (participatory interviews, case studies, and contextual observation methods in the demand exploration phase) and quantitative research (experimental methods and the Mini-Mental State Examination (MMSE) and Barthel Index questionnaires in the product verification phase). This study adopted a four-stage service design: demand exploration, demand definition, design implementation, and product verification. In the stage of demand exploration, 14 elderly people with mild or moderate dementia were interviewed, and two cases were selected for two in-depth observations of horticultural treatment activities. Common obstacles and potential demand points were listed after integration: (1) The safety of elderly patients with dementia can be improved by employing horticultural treatment activities transferred from outdoors to indoors; (2) the objects and facilities used in horticultural activities should be improved to reduce the attention burden of elderly patients with dementia; (3) the elements of reminiscence or familiarity of the mentally handicapped elderly should be increased; (4) the process of gardening and planting can be used by two or four people to improve social and language skills. According to this study, an indoor gardening planting table game was developed. This game includes a group of flower combination prompt cards (including five flower groups: camellia, cherry blossom, chrysanthemum, kapok, and lotus), a group of color and number prompt rings, and a flower base, which provides planting of up to 25 flowers and is matched with the number prompt color rings; then, the combined flowers are planted into the base. In the final experience experiment, 7 participants with free movement of the upper limbs and mild or moderate dementia were selected by the MMSE and Barthel Index to participate in a 5-week experiment. After using a combination of progressive low-level, medium-level, and high-level flower combination tasks, the results showed that the overall performance of the elderly patients with mild or moderate dementia in the MMSE test was improved by the indoor gardening planting table game. However, the treatment effect-size presented a low effect magnitude.

## 1. Introduction

The proportion of the elderly over 65 in Taiwan’s total population increased from 7.10% in 1983 to 14.05% in 2018. It is predicted that the elderly population will exceed 20% in eight years (2026), and Taiwan will be listed as a “super aging society” together with Japan, South Korea, Singapore, and some European countries. In Taiwan, Jiayi County accounts for 18.61% of the elderly population, followed by Yunlin County (17.69%). At the end of August 2017, the number of people over 65 years old in Yunlin County was 119,927, the prevalence of dementia was 8.4%, and the estimated number of people over 65 years old in Yunlin County was 10,073 [1]. Yunlin county is the main agricultural production county in Taiwan, with an agricultural employment population of nearly 40%.

Dementia is caused by brain neuropathy and atrophy, leading to a gradual deterioration of overall cognitive function. The course of dementia can be divided into five periods: mild cognitive impairment, mild, moderate, moderate to severe, and severe. In addition to conscious memory deterioration, attention disorder is also a common symptom of mild cognitive impairment. Before the degradation of language and visual spatial ability, mild cognitive impairment will produce disease-based attention degradation [2], which can lead to restlessness, depression, attempted suicide, and other psychological diseases. Every year, about 9% to 25% of patients with depression who are suspected of dementia develop senile dementia without the proper care [3]. Wu and Lin [4] noted that if people with dementia promptly view the world from a positive perspective, such as focusing on what they have rather than what they lack, they may be free from worrying about what they should do now, which can reduce their restlessness and conflicts with others. Without proper attention, there is no discrimination, learning, or memory. Thus, attention is a key factor in accomplishing tasks or achieving learning outcomes. Therefore, this study believes that attention represents the most important treatment area in the process of dementia, as it promotes the abilities of memory, language, problem-solving, and so on [5].

In a previous study that examined the treatment of non-drug therapy, Söderback et al. [6] found that horticultural activities can effectively enhance the overall cognitive functions of the elderly with dementia. Colorful and flavored plants (such as vanilla) were used to stimulate visual and olfactory sensory perception, and edible plants were used to enhance taste perception. Through horticultural treatment activities, patients with dementia can be provided with opportunities for physical movement. Simple actions can enhance their physiological functions, improve the quality of their sleep at night, and effectively slow down their restless behavior [7]. In addition, after the elderly with moderate dementia participate in horticultural treatment activities, their concentration and activity participation are improved [8], which shows that horticultural treatment has a positive effect on people with dementia. Moreover, horticultural therapy can also improve the focus of elderly patients with dementia. Participation in social activities also has positive results and improve a patient’s focus on daily life [9], which will further strengthen his or her communication and cognitive abilities [10,11]. Therefore, horticultural therapy can not only improve patients’ overall cognition but also improve their expectations and active participation, as well as enhance their psychological self-confidence. From the perspective of social behavior, horticultural therapy improves interpersonal relationships and cultivates interaction and coordination with other people by encouraging patients to care about planting and growth and increasing their fun in life. Therefore, horticultural activities have positive effects in the daily lives and social functions of the elderly with dementia [12].

Consequently, this study seeks to improve the design of related products, such as equipment, teaching aids, and assistive devices, used in horticultural treatment activities [13]. For example, in the design of related implements and plants, plants or auxiliary tools are used to emphasize changes in color and touch [14]. Color and touch can stimulate the sensory cognition and memory of patients with dementia [15], which can enhance their attention. Using familiar cognitive memories will make patients feel comfortable and in control [16].

The purpose of this study is to explore the potential needs and deficiencies of horticultural therapy activities in existing institutions, and to further develop indoor horticultural table games that can improve the overall cognitive function of the elderly with mild and moderate dementia. This study adopts the service experience insight method for its service design, with user experience as the core. By using the familiar horticultural activities of residents in Yunlin County, the elderly with mild, and moderate dementia in Yunlin County were the core users. This service experience insight will help us understand users’ lifestyles and behavioral patterns, as well as their real needs and barriers [17]. For example, Tseng [18] used service to experience insights and discovered the potential social needs of elderly people living alone. Duan [19] also used this method to shape the needs of remote health care services for retired elderly people. Using this method, Xiao and other scholars [20] found that care and social workers encountered potential defects and obstacles when they used a Professional Care Management System. This study applies the contextual inquiry from the theory of Service Design at a private elderly care center in Douliou city, Yunlin County, Taiwan. The data analysis and behavioral modeling resulted in an improved understanding of the cognitive impairment of elderly patients with dementia and revealed the obstacles and needs of the elderly via horticultural treatment activities. According to these results, a design policy of indoor horticultural activities is proposed to improve the overall cognitive function and awareness of elderly patients with dementia.

## 2. Method

The service design flow and tools used in this study comprise four kinds of service design processes and tools commonly used in the industry: IDEO (an international design company in the U.S.) Design Process [21], the Double Diamond Design Process (Design Council 2008), Service Experience Engineering [22], and the Idea Service Design Process [23]. The research method in this study was divided into four stages of service design: Demand exploration, Demand definition, Design execution, and Product verification [18], as shown in Figure 1. The research method used in this study was a combination of qualitative research and quantitative research and was integrated into the four service design stages to develop indoor gardening table games to improve the overall cognitive functions of the elderly with dementia in Yunlin County. (1) In the demand exploration stage, qualitative research was the main method used, including semi-structured interviews, case studies, and contextual experience insights, assisted by the Mini-Mental State Examination (MMSE) and Barthel index (assessing functional independence) to select target cases, which were used as contextual experience observation cases during horticultural treatment activities. (2) In the demand definition stage, the data collected in the previous stage was used to form a focus group with three cross-domain experts to summarize the potential demands by the affinity graph method and to form a product design policy by integrating, discovering, and exposing the opportunity points. (3) In the design and development stage, these requirements and opportunities became the focus of design service improvement. (4) In the design verification stage, quantitative research, the indoor gardening table game experimental design, and the MMSE questionnaire were used to confirm whether the product could improve the overall cognitive function of patients with dementia.

All participants gave their informed consent for inclusion before they participated in this study. The study was conducted in accordance with the Declaration of Helsinki, and the protocol was approved by the Research Ethics Committee of National Changhua University of Education (NCUEREC-107-044), 29th October 2018.

### 2.1. Demand Exploration

#### 2.1.1. Semi Structured Interview

In this stage, elderly people and caregivers who were willing to be interviewed were recruited from a Douliu City care center. The interview content was first based mainly on basic data, while case introductions, nursing evaluation, etc., were added later. Finally, the physical condition and cognitive performance of the elderly people were determined by using the Barthel Index (provided by the care center) and the Mini-Mental State Examination (MMSE). There were 14 participants (3 males and 11 females), with an average age of 82. Each participant had one semi-structured interview and an MMSE test; the target group of this experiment comprised 14 interviewees. Each interview lasted about 30 min on average, referred to Appendix A. We sought to determine whether the participants had an agricultural background, whether their upper limb strength was used freely, and their degree of dementia (MMSE score) and enthusiasm for participating in gardening activities.

#### 2.1.2. Contextual Inquiry

In this study, the participatory observation method was used for natural integration into the study group, and the two selected cases were observed. Through the two horticultural treatment activities in the conservation institution, the observation and interview objectives of the context investigation experiments were taken. By examining five aspects of the activities (environment, interaction, objects, and user), we observed and collected case behavior in horticultural treatment activities and interaction behavior and information with the environment, objects, and stakeholders.

### 2.2. Demand Definition

In this stage, the affinity graph method was applied to summarize the potential demands of users. This method originated from Kawakita Jiro, an anthropologist, in 1965. This method was conceived to improve the data processing ability and results of field surveys [24].

### 2.3. Design Implementation

According to the research results of the demand definition stage, this study established the product system architecture and design objectives. The hardware was an indoor gardening planting game, which aimed to increase the stimulation of the five senses of the elderly, strengthen their concentration and selective cognition, and improve their cognitive performance and attention step by step according to low, medium, and high levels of gardening planting tasks.

### 2.4. Product Verification

The purpose of the product validation phase was to test whether the indoor gardening planting table game could improve the cognitive performance of the elderly with mild and moderate dementia. This verification was divided into two steps: the test before and after the test of overall cognitive performance, and the experiential experiment of the indoor gardening planting table game. MMSE was used for the pre- and post-cognitive performance.

#### 2.4.1. Participants

The participants were recruited from another daycare center in Doulie, according to patients’ persona characteristics. After the MMSE test and the Barthel Index, there were 20 people chosen with mild and moderate dementia, and 10 people with slight dependence on their body functions (with full function of their upper limbs). After cross analysis, there were 7 remaining people with mild to moderate dementia and upper limb extremity mobility—four men and three women. The average age of the patients was 86 years old. Four were primary school graduates, and 3 were secondary school graduates. All of them were retired people engaged in agriculture. All participants and their families signed the informed consent form and received an allowance for participating in the experiment.

#### 2.4.2. Apparatus, Materials, and Procedure

This experiment was divided into two parts: (1) The pre-test and post-test of cognitive performance and cognitive attention; (2) The experiential experiment of the horticultural treatment table game.

Before the experiment, the experimenter first conducted 5 min of instruction on the experiment, and then the participants were examined by the MMSE test to confirm their cognitive performance and attention score before the table game; the test time was about 15 min.The experiment using gardening table games was conducted at a daycare center in Douliu City for 5 weeks. There were 9 flower combination tasks. According to the difficulty, these tasks were divided into low, medium, and high-level stages, referred to Appendix A. Each stage involved 3 combinations of tasks and 2 tasks per week, Figure 2.

### 2.5. Data Analysis

The research method of this paper was divided into four stages: demand exploration, demand definition, design implementation, and product verification. In addition to the stage of design execution, which was mainly product design, the stage of demand exploration and demand definition involved qualitative data analysis, and the stage of product verification involved quantitative data analysis.

Demand exploration:
Persona: In this study, we use the method of persona analysis to select the representative participants [25] based on the data collected from the semi-structured interview and the common behaviors and characteristics of the interviewees. In this study, two participants (1 elderly patient with mild dementia and 1 elderly patient with moderate dementia) were selected; both patients could easily use their upper limbs (i.e., their Barthel Index was mild), which were represented as Case-1 and Case-2, respectively.Behavior Modeling analysis: Based on the observation and interview of the above two selected cases, this study collected the interactive behaviors and information of the cases using horticultural therapy activities based on the five dimensions of activities, environment, interactions, objects, and the user (A,E,I,O,U). In constructing the behavioral models [26], four working models were produced: an interactive model, a sequential model, a utensil model, and an entity environment model. (1) Interaction model: The interaction model can simplify the way that users interact with the environment and others into a more easy-to-understand model. In this study, an interaction model of two cases with other people and fields in horticultural treatment activities was produced, and the obstacles to the interaction process are shown. (2) Sequential model: the behavioral tasks in the interactive model are listed in chronological order, and the contact points are clearly presented. In this study, all the steps in the horticultural activities were performed in a chronological order to determine the problems and repetitive behaviors that emerged in the process of the behavior and to elaborate on those obstacles. (3) Object model: Through the object model, we can see how the user employs objects to complete the specific tasks during the horticultural treatment. In this study, the objects used by the elderly in the two periods were photographed for an object model analysis, and the obstacles were listed. (4) Entity model: This is the field where users engaged in the horticultural activities. This model describes the layout and structure of horticultural activities, the purpose, the method being used, and the activity route of users in the environment, as well as various related artifacts.
Demand definition: In this stage, the affinity graph method was applied to summarize the potential demands of users [24]. In this stage, 3 experts, including nurses, designers, and dementia specialists, were recruited to form a focus group to define the potential needs of horticultural treatment through affinity mapping.Product verification: In the product verification stage, a paired-sample T-test and repeated measure design for the dependent samples were used to test whether the overall cognitive function of the elderly with mild to moderate dementia improved before and after the horticultural table game experiment; the effect size was also tested. The overall cognitive function test used the MMSE. In this experiment, the pre-test and post-test scores are tested for normal distribution. This paper hypothesizes that the average score of the participants’ overall cognitive functions after experiencing the horticultural treatment table game experiment will be higher than that of the participants who did not experience the horticultural table games.

## 3. Results

The purpose of this study is to improve the cognitive performance of patients with dementia. The results are presented in four stages: (1) Demand exploration stage, (2) Demand definition stage, (3) Design implementation stage, and (4) Product verification stage.

### 3.1. Demand Exploration

#### 3.1.1. The Persona for Horticultural Treatment

In this study, the participants were mainly mild and moderate dementia patients, and their physical and mental functions were also mildly dependent; that is, the patients could use their upper limbs and lower limbs freely. The Barthel Index (provided by the care center) and MMSE scale were applied to 14 mentally handicapped elderly people in a care center. After a cross-analysis of the two tests, we selected two main patients with mild physical dependence and mild–moderate mental impairment as the representatives for the study. The patients were named Case-1 and Case-2. The representative patients were selected according to the following four classifiers: (1) agricultural background; (2) the ability to use their upper limb strength; (3) an attention cognitive score lower than average; and (4) enthusiasm to participate in the activities.

#### 3.1.2. Contextual Inquiry Observation

We conducted an in-depth study on the service needs of these two elderly patients in the two stages of horticultural treatment activities. The first stage was “Happy Mother’s Day”, and the second stage was “All kinds of sweet potato leaves”. Based on the five dimensions of activity, environment, interaction, object, and user, this study analyzed all the internal operational and process details of the horticultural activities, such as the environment, tools, participants, etc., and then presented the obstacles to the elderly patients with dementia for horticultural treatment activities. In the user demand part, we observed physical function degradation, the need to be accompanied, past experiences, etc.; there were also obstacles and unmet needs in the horticultural treatment. For the obstacle and demand portions of the objects, the horticultural utensils needed to have indication signs, such as color or intuitive mechanism design configurations, as well as appropriate table and chair height configurations, to meet the needs of the elderly patients with dementia and to avoid obstacles. Finally, in environmental facilities and field planning, wheelchair users had to be considered. Outdoor planting activities were prone to accidents, and it was not easy to ensure everyone’s safety, increasing the possibility of issues.

#### 3.1.3. Behavioral Modeling

This study collected data from interviews and observations using contextual inquiry and integrated the behavioral models of the two selected elderly patients with dementia to produce a convergence interaction, sequence, object model, and physical environment model to determine the common obstacles and demand points.

Interactive Model: The elderly patients with dementia who engaged in the horticultural treatment activities, as well as people, matter, and thing interactions, have been illustrated. Whether the event was held indoors or outdoors, the environment was always a common obstacle in the implementation of the activities. For example, there were many wheelchairs, dinnerware implements, furniture, etc., in the room, which made it difficult to push the wheelchair. Outdoors, there were weather factors and the risk of losing patients. When communicating with the patients, the involvement of the department head and social workers made the elderly patients with dementia more likely to concentrate on flower arranging during the flower arranging activities, as shown in Figure 3.Sequence Model: During the processes of flower arrangement and planting sweet potato leaves by the elderly with dementia, we found that if the planting activities were carried out according to the past life experiences of the elderly patient, the participation of the patient improved. For example, when participant 1 learned that the activity of planting potato leaves was to be held outdoors, he walked outside to teach everyone how to grow sweet potato leaves. This study found that each plant had its own planning process, as shown in Figure 4.Object Model: This study found that common obstacles indoors and outdoors included the following: (1) Flower arrangement was affected by red triangles and white circular stickers. When there were plastic stickers on the sponges, the elderly patients focused on and inserted the hole first. (2) Plastic chairs and benches were used as simple tables (about 45 cm high) in both horticultural therapy activities, which resulted in difficulties in flower arrangement for the patients participating in the activities. (3) When arranging flowers, some elders would stop without knowing how to arrange the flowers, Figure 5.Environment Models: Through this map, we found that the elderly patients with dementia had limited movement lines when they engaged in indoor gardening activities because of too much debris and too many wheelchairs, as seen Figure 6.

### 3.2. Demand Definition

This research studied the two horticultural therapy activities of the care center. The first event was “Happy Mother’s Day”, and the second was “All kinds of sweet potato leaves”. After several contextual inquiries, the horticultural activities were analyzed with the five facets of A.E.I.O.U. (activities, environments, interactions, objects, and users), and the key sentences from the observations and interviews were extracted and classified into seven types. These types can be combined into three major categories: people, appliances, and the environment. The users’ behavioral patterns and potential needs were expressed with an affinity graph. Finally, the service requirements and opportunities were consolidated as “indoor tabletop horticultural treatment planting activities”. We offer four recommendations for constructing a service product system aimed at improving elderly attention and cognition:The safety of the elderly with dementia can be improved by transferring horticultural treatment activities from outside to inside: Elderly patients with dementia are generally unable to stay outdoors for a long period of time because of the deterioration of their upper and lower limbs and their high risk of accidents. The elderly in the care center were almost all seated in wheelchairs, causing frequent collisions with other patients while moving. Therefore, the study was proposed to move from outdoors to indoors, and the activities were changed to tabletop gardening. Although the elderly patients were not as physically fit as they once were, they still enjoyed the fun of planting in a wheelchair (Table 1).The objects and facilities used in horticultural activities can be improved to reduce the attention burden of the elderly with dementia: Based on the suggestions from interviews with functional therapists, horticultural therapy could be added to the Color Trails Test [27,28]. Participants could train their attention according to order, number, and color cues, providing a quick and effective way to test patients with a low educational level, those unfamiliar with English letters, or those with reading or language disorders. In this study, multiple sensory stimuli from color or sound cues were added to the start of the flower arrangement to train concentration and selective attention, in order to achieve a rehabilitation effect, as shown in Table 1.Experiences or sounds that are familiar or reminiscent to the elderly can be added: Familiar or reminiscence sounds or experiences can promote awareness of the elderly with dementia. Sounds should be chosen according to the interests of the elderly, such as the sound of a bicycle, the sound of a train whistle related to previous job experience, and other sounds related to their living environment. Moreover, nostalgic experiences related to farming, such as planting flowers, plants, fruits, vegetables, rice, etc., can stimulate the elderly with dementia to improve their cognitive functions. Accordingly, the tabletop horticultural treatment activities were designed with sounds based on the previous experiences of the elderly, in order to promote thinking, participation, and to focus their attention when handling the products (Table 1).The processes of gardening and planting can be used by two or four people to improve social and language skills. Two major issues emerged from the five observational perspective of A.E.I.O.U. design thinking. First, most of the elderly dementia patients in the care center generally meet with their families on the weekends. Second, after the intervention of the head of the care center, the elderly showed more facial expressions in their interactions, and even took the initiative to answer questions actively. This study suggests that the concept of long-distance companionship can be added to make the elderly feel the companionship of their family members. For example, planting games can instantly reflect the faces or voices of family members. Families can offer care and encouragement by using the application software of smart phones. Promoting emotional interactions can also enhance attention awareness and make the elderly more enthusiastic about engaging in planting games (Table 2).

### 3.3. Design Implementation

According to the product system architecture and product design guidelines from Table 2, this product design was divided into two categories: hardware and software. The hardware part of the design included the body of the indoor table planting game and the internal sensor and cloud server. The purpose of this design was to increase the hand-eye coordination of the elderly and strengthen their cognitive performance. The software part of the study included the mobile app program design and interface design for family members or caregivers. Based on this, the prototype design for the indoor table planting game was divided into two stages. The first stage was primarily used to strengthen cognitive performance. In the design, nostalgic elements were combined, sound and visual aids were added, numbers and color rings were added to the base of the gardening plants, and gradual game training was combined to improve the cognitive performance of the elderly with mild to moderate dementia. In the second stage, sensors were placed in the flowers and bases to increase auditory and visual stimulation, strengthen the training of attention, record the duration, accuracy, and number of reminders of the game training process, and set up the design and production of the cloud server and app. At the same time, we added the concept of long-distance companion care (Table 2).

The prototype design developed in this study was based on stage one. First, we needed to confirm the role of this gardening table game in improving attention, and then we carried out stage two’s hardware development. The table game of horticultural planting developed in this experiment included a set of flower combination prompt cards (including five flower combination modes) and five kinds of flowers (camellia, cherry blossom, chrysanthemum, kapok, and lotus); each part of the flower consisted of petals, bracts, leaves, stems, etc., and included a base that could have 20 flowers inserted into it, as well as a color ring with digital prompt. The following describes the design concept of each functional accessory, referred to Table 2.

Five kinds of flower cards: These cards presented the composition of the flowers, including five kinds of flowers, camellia, cherry blossom, chrysanthemum, kapok, and lotus, referred to in Table 3, to prompt the composition of flowers. By reading aloud, the elderly with dementia were able to extract the relevant information of flower composition from memory.Five kinds of flowers were used (camellia, cherry blossom, chrysanthemum, kapok, and lotus), including their petals, bracts, leaves, and stems for all parts of the flowers. Based on the farming experience of people who live in the Yunlin County of Taiwan, the five most common flowers in Taiwan in the four seasons were extracted and bloomed according to different seasons.Color and number prompt ring: With the help of numbers and colors, users could be trained to improve their concentration and selective attention, as well as their divided attention. Referring to the Color Trails Test [27,28], the elderly patients were asked to insert the flowers into the base according to their numerical order and color.In the future, sensors and light-emitting elements could be installed on the base to record the user’s duration, accuracy, and number of reminders and upload those data to the cloud for family members and caregivers to track their improvement.

#### Prototyping

Because a large number of product components were required, we decided to make these components with 3D printing. Twenty-five sets of petals, flowers, and stems and leaves were produced according to the Color Trails Test (CTT). Besides red, yellow, and blue colors, other colors were added for sharper contrast. According to the study of Huang and Chiou (2016) [29], green, yellow, and blue offer a sense of security and intimacy for the elderly while the most comfortable colors found in nature are green, orange, and blue. In summary, this study used a color system of red, yellow, and blue to enhance the selective attention of the elderly. The base was made of acrylic and PLA plastic via a local 3D printer, as shown in Figure 7.

### 3.4. The Results of Product Verification

#### Results of the Mini-Mental State Examination (MMSE)

In this study, we hypothesized that the elderly patients with mild to moderate dementia would improve their overall cognitive performance after engaging in the indoor horticultural planting table games and that the post-test MMSE score would surpass the pre-test score. A paired-sample t-test and the repeated measure design of dependent samples were used to test whether the overall cognitive functions of the elderly patients with mild to moderate dementia improved before and after the horticultural table game experiment; the effect size of the average pre-test and post-test score was also tested.

According to the Kolmogorov–Smirnov and Shapiro–Wilk tests, the MMSE scores measured before and after were not significant and are expressed according to their normality in Table 3.

As seen in Table 4, the mean pre-test and post-test MMSE scores were 18.57 and 20, respectively, and the difference between them was 1.43. The post-test and pre-test MMSE scores were strongly and positively correlated (r = 0.97). There was also a significant average difference between the post-test and pre-test MMSE scores (t = 3.33, *p* = 0.016). The statistical results showed that the cognitive performance of users in the MMSE test improved after using the indoor horticultural table games. Further, the effect sizes of the pre-test and post-test mean scores was tested, including Cohen’s d = −0.33, with an Effect-Size = 0.2. An Effect-Size (ES) of 0.2 indicates that the mean of the post-test is at the 54th percentile of the pre-test, with a nonoverlap of 14.7% in the two distributions, which represents a low effect magnitude [30].

## 4. Discussion

In this study, the four stages of service design (demand exploration, demand definition, design execution, and product validation) were undertaken to identify potentially unmet needs and deficiencies in existing horticultural activities and to improve the overall cognitive functions of elderly patients with mild and moderate dementia in Taiwan’s Yunlin County. The high proportion of elderly people in Yunlin County, Taiwan, with a prevalence of dementia (8.4%) indicates that Yunlin County contains 10,073 people with dementia over 65 years of age. Yunlin county is the main agricultural production county in Taiwan, with an agricultural employment population of nearly 40%. It is an agricultural county with a large area and a small population. The residents live in scattered villages around a large area of farmland, making is much more difficult to get medical treatment and care for dementia than in the metropolitan area. Therefore, this study focused on the most familiar horticultural activities for elderly patients with dementia in Yunlin County. The main purpose of this study was to explore the potential needs and deficiencies of horticultural treatment activities in existing institutions and to further develop indoor horticultural table games that can improve the cognitive function of elderly with dementia.

### 4.1. Potential Demands for Elderly Patients with Mild and Moderate Dementia

The purpose of this study was to use the context inquiry of service design to collect elderly behavioral problems in indoor horticultural treatment activities and interview caregivers, in order to define the main objectives of this study through the help of personas. Based on the results of these interviews, the interaction, sequence, tools and objects, and physical environmental model were integrated, and then the affinity graph method was used to converge the collected information to determine the potential needs of the elderly with dementia in a Douliu private elderly care center of Yunlin county to improve their cognition and attention in indoor horticultural treatment activities. The following are the potential requirements for consolidation: (1) horticultural treatment activities can be carried out indoors; (2) reminiscent elements can be added; (3) this exercise can be used to improve patients’ concentration and selective attention; (4) it is better for patients to have the company of acquaintances; (5) an intuitive operation design is important; (6) a sitting position or wheelchair can also be used; (7) caregivers must consider environmental friendliness for patients. These seven items were integrated into the categories of people, tools, and the environment. The overall service needs and opportunities are as follows: (1) the safety of the elderly patients with dementia can be improved by moving the horticultural treatment activities from outdoors to indoors; (2) improving the objects and facilities used in horticultural activities can reduce the attention burden of the elderly with dementia; (3) adding familiar or reminiscent experiences or sounds can help the elderly; (4) the process of gardening and planting can be used by two or four people to improve social and language skills.

It is not surprising that the above conclusions were drawn during the exploration and definition stage. Gardening is one of the favorite activities of the elderly with dementia in the care center, but it must be done outdoors because of the need for planting. However, due to the funding and safety reasons of each care center, space and equipment are limited, so such activities are rarely engaged in. Therefore, this study changed outdoor gardening activities into indoor games through design improvement, which can be used by general care institutions and families.

The second observational trigger point was that the elderly with dementia were often unable to focus or concentrate their attention when planting. For example, when arranging flowers, the elderly often felt frustrated because they did not know where to insert the flowers, which made the activities not run smoothly or become stagnant; the elderly with dementia were not confident and could not focus on the activities. However, how can we make these elders focus and maintain their concentration? This product uses the experience of farming in Yunlin County to add a reminiscence element to the horticultural products. Using the past farming experiences of the elderly with dementia [31], the common flowers of Taiwan’s four seasons (camellia, cherry blossom, lotus, chrysanthemum, and kapok) were added to the table game of gardening and planting. Based on the familiar composition of these flowers’ growth patterns, and with the help of color and voice reading, the elderly patients with dementia were able to easily complete the tasks of the table game.

### 4.2. Improvement of Cognitive Performance on MMSE

The average score of the MMSE in the post-test was statistically higher than that in the pre-test after the elderly patients with mild and moderate dementia engaged with the horticultural planting products. However, the effect-size represented a low effect magnitude. This showed that the horticultural products and the operation mode of the experiment were helpful in improving the cognitive functions of dementia among the elderly patients with mild and moderate impairment of their functions. However, due to the small number of samples and the lack of a control group in the experimental design, the improvement effect of the treatment was reduced.

The improvement of these scores in the pre-test and post-test proved that the method of reading aloud [32], the reminiscence elements [31], and the color number ring of the Color Trail Test [27,28] were helpful. In the experiment, when the flower chart card was presented, the subject was asked to vocally describe its components, select the corresponding flower components, and vocally express the combination process, in order to facilitate concentration during the tasks. Likewise, the gardening planting table game used visual aids (picture cards) combined with color number rings, which also helped the subjects focus easily and improve their concentration. In addition, this experimental operation adopted progressive learning therapy [33], which was understood by the examinees, so that the gardening planting game could better improve cognitive functions.

### 4.3. Research Limitations

In this study, user experience (as the core of service design) was used to determine the barriers and needs of the elderly people with dementia when they engaged in horticultural treatment activities. Thus, we designed a horticultural planting table game to improve cognitive attention. Through MMSE verification, after 5 weeks of product experience, as well as initial, intermediate, and advanced learning treatment, there were statistically significant differences between the pre-test and post-test values in cognitive function. However, there was only one horticultural planting product included, resulting in small experimental samples. Further, there was no control group in the experimental design, which reduced the improvement effect of the treatment.

## 5. Conclusions

We used a four-stage service design process (demand exploration, demand definition, design execution, and product verification) to explore the potential needs and expectations of elderly patients with mild and moderate dementia by using existing horticultural activities. In the demand exploration stage, this study aimed to collect the real behavior and satisfaction needs of users and stakeholders in horticultural activities from five perspectives: activity, environment, interaction mode, use objects, and each related person and user. During the stage of demand definition, this paper put forward the specific product system architecture and design specifications and developed indoor gardening planting table games. According to the needs of the care center, the outdoor horticultural activities were changed to “indoor table planting therapy activities”. Even if the physical strength of the elderly patient is poor, the patient can still enjoy the fun of planting in an indoor wheelchair. This type of “indoor table planting game” is mainly used to strengthen the training of cognitive attention. The game’s design combines reminiscence elements, audio reading, and visual aids; based on horticultural planting, it is also supplemented with hints of color and digital rings, combined with progressive game training, in order to improve the cognitive attention of the mentally handicapped elderly.

Finally, after the experiment, the indoor gardening planting game could, in the future, be equipped with sensors and light-emitting elements on the base, which can record the user’s operating duration, accuracy, and number of reminders and then upload that information to the cloud. These data could then be used by family members and caregivers to track improvement. Although this product is mainly used by elderly patients with dementia in the Yunlin County care center, it can also be used in other nursing institutions.

## 6. Patents

The indoor garden planning table game developed in this study has obtained a new patent of the Republic of China, with the certificate number of m579545 and the creators, Winger Sei-Wo Tseng, and Yi-te Yeh.

## Figures and Tables

**Figure 1 ijerph-17-01483-f001:**
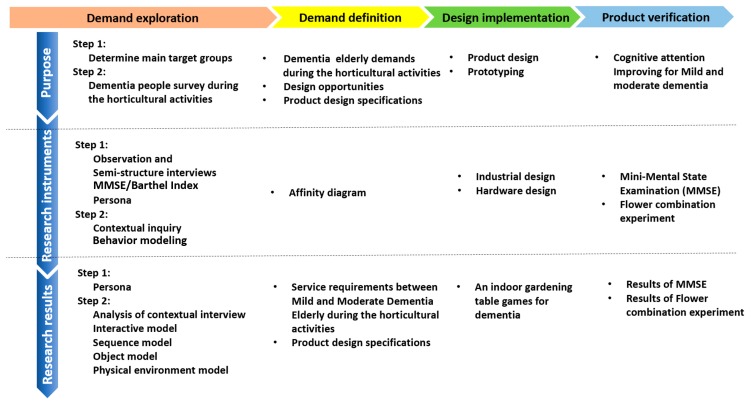
The research design process used in this study, including demand exploration, demand definition, design implementation, and product verification.

**Figure 2 ijerph-17-01483-f002:**
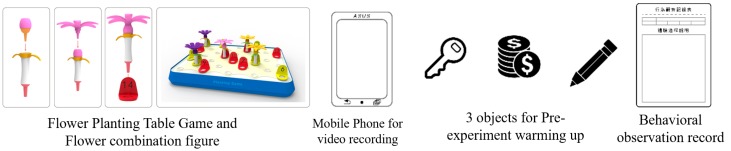
Equipment used in this research experiment.

**Figure 3 ijerph-17-01483-f003:**
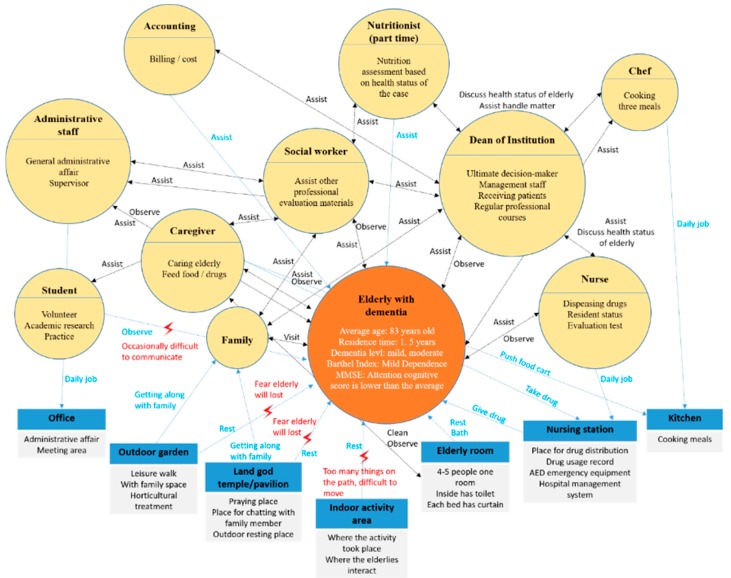
Interactive model.

**Figure 4 ijerph-17-01483-f004:**
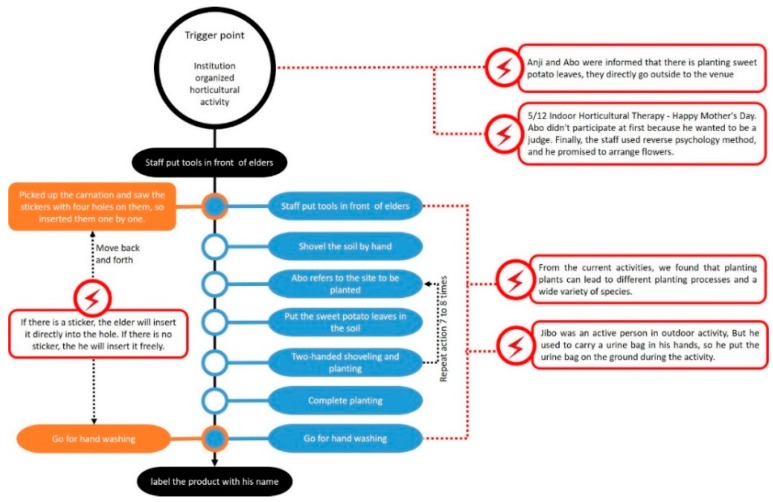
Sequence Model.

**Figure 5 ijerph-17-01483-f005:**
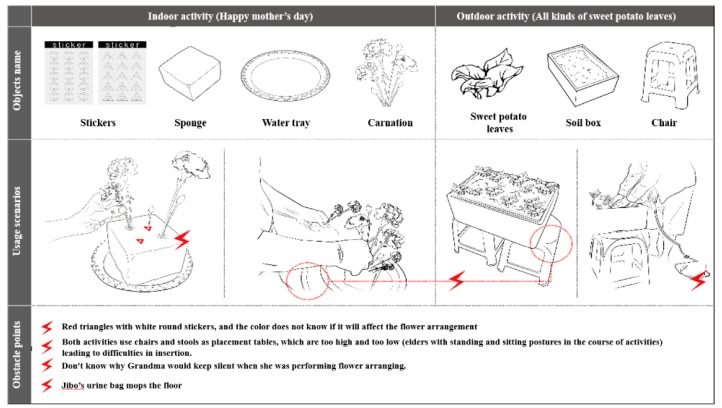
Object Model.

**Figure 6 ijerph-17-01483-f006:**
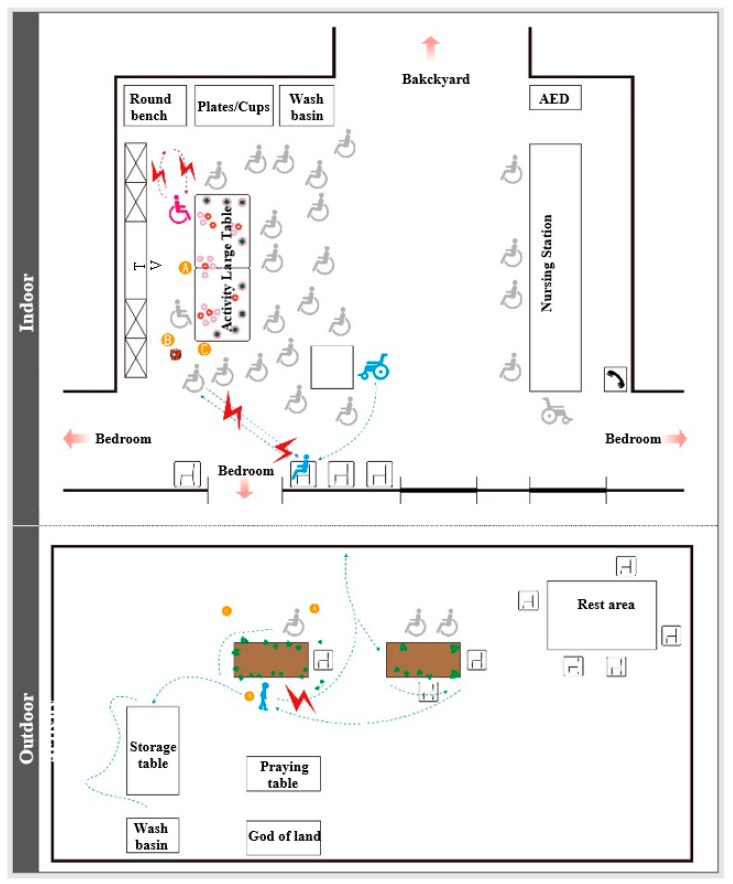
Environmental Model.

**Figure 7 ijerph-17-01483-f007:**
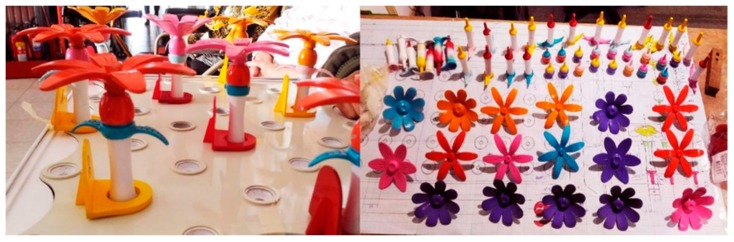
Model making process.

**Table 1 ijerph-17-01483-t001:** The design guidelines of the indoor garden table game for improving the cognitive attention of elderly dementia patients.

Design Target		Function	Requirements
User	Elderly patients Mild/Moderate Dementia	An indoor planting game to improve the cognitive attention of the elderly with mild to moderate dementia	Using contrasting colors to improve attention cognition
			Add nostalgia to the product design
			Adding sound and visual aids for cognitive attention
			Transmitting usage records and analyzing usage status through an app device
Design object	Indoor dementia attention rehabilitation game equipment	An indoor table game to improve the cognitive attention of the elderly with mild/moderate dementia	Add nostalgic elements: Use Yunlin residents’ farming experience to bring flowers from different seasons/places into the game.
			Add sound, number, or color cues to train users to improve their concentration and selective attention
			Add the flower structure to users’ oral repetition to extract relevant information from memory
			Use number and color combinations to train user’s choice/shifting attention
		Design a sensor device	Transmit game response time through stress sensing
Server		Record, analyze, and transmit events to the cloud	Record, analyze, transmit reaction time, and score
Cloud		Send events to smart devices	

**Table 2 ijerph-17-01483-t002:** The prototype design of the indoor gardening planting table game for improving the cognitive attention of mild and moderate dementia elders.

Functional Accessory	Prototype Design
Flower combination tip card	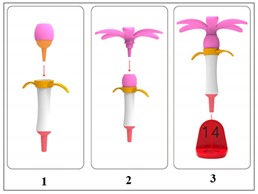
Five kinds of flowers: camellia, cherry blossom, chrysanthemum, kapok, and lotus	Camellia	Kapok	Mountain cherry blossom	Chrysanthemum	Lotus
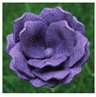	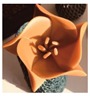	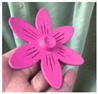	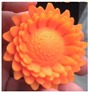	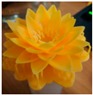
Color and number plate	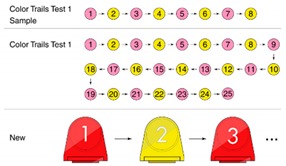
Flower base	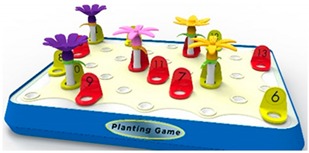

**Table 3 ijerph-17-01483-t003:** Tests of normality.

Category	Kolmogorov–Smirnov ^a^			Shapiro–Wilk		
	Statistic	df	Sig.	Statistic	df	Sig.
Pre-test	0.129	7	0.200 *	0.963	7	0.843
Post-test	0.169	7	0.200 *	0.964	7	0.851

*. This is a lower bound of the true significance. ^a^. Lilliefors Significance Correction

**Table 4 ijerph-17-01483-t004:** The T-test for the difference between the mean post-test and pre-test the Mini-Mental State Examination (MMSE) score for the Indoor gardening table game.

	Pre-Test	Post-Test	t	df	Sig. (2-Tailed)
	Mean	Mean			
**TOTAL**	18.57	20.0	3.33	6	0.016

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
