# Peer review of "An Indoor Gardening Planting Table Game Design to Improve the Cognitive Performance of the Elderly with Mild and Moderate Dementia"

_ijerph, 2020, doi:10.3390/ijerph17051483_

Round 1
Reviewer 1 Report
Dear authors,
thank you for the opportunity to read your work. The problem you are investigating is undoubtedly important, and every method that seeks to help the patients with cognitive impairment and dementia is worthy of appreciation and research. However, I see serious shortcomings in your work (or in this version of its presentation), including the imprecision of the terms used; multiple repetitions of text and redundant verbosity in introduction, methods section and discussions; lack of clarity in description of methods and conclusions; questionable and, in my opinion, inadequate assessment of cognitive function based solely on the MMSE questionnaire; too small sample of patients.
Generally the introduction is a bit too long and could be shortened without losing clarity and focusing on the main idea of the work.
The literature review should provide a brief overview of previous research by other authors, not starting to discuss methodological aspects of this work.
You should avoid the repetitions of the text and redundant verbosity in introduction, methods section and discussions (for example lines 47-50 and 134-137).
MMSE is a common standard of short cognitive assessment and does not need additional explanation and description. However, it has well-known limitations and usually is insufficient to assess changes in cognitive function in dementia clinical trials. Your presented cut-off values for MMSE valuation (lines 324-328; 18-24 points are mild dementia, 16-17 points are moderate dementia, and less than or equal to 15 points are severe dementia) are highly questionable.
Conclusions should be clearer, shorter, and reflect specific findings of your work (not declarative general statements).
I am sure that the presentation of your important work can be significantly improved by a substantial revision of the article.
Author Response
Response to Reviewer 1 Comments
We are very grateful to the reviewer1’s comments which make the results of this paper more solid. The manuscript was revised in accordance with the reviewers' comments and suggestions, followed by professional extensive English editing (English language editing by MDP, English-16242, Attachment 1). The followings are our descriptions of revision according to the reviewer1’s comments.
Point 1: Thank you for the opportunity to read your work. The problem you are investigating is undoubtedly important, and every method that seeks to help patients with cognitive impairment and dementia is worthy of appreciation and research. However, I see serious shortcomings in your work (or in this version of its presentation), including the imprecision of the terms used; multiple repetitions of text and redundant verbosity in introduction, methods section, and discussions; lack of clarity in description of methods and conclusions; questionable and, in my opinion, inadequate assessment of cognitive function based solely on the MMSE questionnaire; too small sample of patients.
Response 1: Thank you very much for your comments. This paper is rewritten and integrates the introduction with the literature discussion. At the same time, the research methods/results/discussions/conclusions are also readjusted.
Introduction: The introduction is rewritten and combined with the literature reviewed section, and the redundant words and sentences are also modified(Please referred to 1. Introduction,Line44-108). Research method: In this paper, we add a Data analysis section to make the method clearer (Please referred to Line 248-292). Discussion and Conclusion section: we also rewrite these two sections, and focus on the important conclusions of this article and delete irrelevant parts. (Please refer to Discussion and Conclusion) MMSE: This study uses MMSE, which is based on the fact that this questionnaire is one of the current evaluations of dementia medication and National Health Insurance in Taiwan, so this study uses the MMSE questionnaire to evaluate. However, more evaluation methods (i.e. CDR) of dementia should be added. Insufficient Samples: This is one of the limitations of this study, so we also analyzed the effect size to confirm its effectiveness and found that the effect is indeed low due to the insufficient number of samples (Please referred to Line 491-494)
Point 2: Generally the introduction is a bit too long and could be shortened without losing clarity and focusing on the main idea of the work.
Response 2: Please refer to Response 1, the introduction has been revised and combined with a literature review (Please referred to 1. Introduction,Line44-108).
Point 3:The literature review should provide a brief overview of previous research by other authors, not starting to discuss methodological aspects of this work.
Response 3: Please refer to Response 1, the introduction has been revised and combined with a literature review (Please referred to 1. Introduction,Line44-108).
Point 4:You should avoid the repetitions of the text and redundant verbosity in the introduction, methods section and discussions (for example lines 47-50 and 134-137).
Response 4: Thank you for your comments. The article has been simplified and revised. Please refer to Response 1.
Point 5: MMSE is a common standard of short cognitive assessment and does not need additional explanation and description. However, it has well-known limitations and usually is insufficient to assess changes in cognitive function in dementia clinical trials. Your presented cut-off values for MMSE valuation (lines 324-328; 18-24 points are mild dementia, 16-17 points are moderate dementia, and less than or equal to 15 points are severe dementia) are highly questionable.
Response 5: Thank you for your comments. For the MMSE dementia evaluation value, this study is based on the evaluation value of the Taiwan National Health Insurance medical dementia evaluation. Please refer to reply 1, idem d.
Point 6: Conclusions should be clearer, shorter, and reflect specific findings of your work (not declarative general statements).
I am sure that the presentation of your important work can be significantly improved by a substantial revision of the article.
Response 6: Thank you for your comments. We have rewritten the conclusion. Please refer to line 577-612.
Attachment 1

Reviewer 2 Report
Patients with dementia often have multiple cognitive disorders such as inattention and memory. Attention is the key factor to complete tasks or learning outcomes. This study focuses on the horticultural treatment activities of the elderly with mild and moderate dementia in Yunlin County, Taiwan.
The purpose of the study was to explore the potential needs and deficiencies of horticultural therapy activities in existing institutions, and to further develop indoor horticultural table games that can improve the attention of the elderly with mild and moderate dementia.
The paper has a clear friendly structure (Introduction, Literature reviews, Method, Results, Discussion, Conclusions) and the subject is interesting and useful as the paper raises important issues regarding the activation and improvement of cognitive abilities of the elderly with mild and moderate dementia.
The manuscript stands carefully developed methodology and good in- depth analysis of the results. The text is supplemented by 6 tables and 7 figures and enriched with 43 appropriate references.
However there are some issues which certainly should be improved:
Major comment
The manuscript is too large and the style is too lengthy. It took me over 2 hours to read the entire article! This makes the text boring and the main idea and goals of the article can be easily lost. So the manuscript should be considerably shortened by at least 1/3 of the length. In addition, there are many unnecessary repetitions in the text.
Comments on individual sections:
Introduction, line 53: word ‘moderate’ is repeated twice Introduction, line 84: word ‘mild’ is repeated twice Literature Reviews: I am not sure if this section is really necessary. Anyway it has to be at least shortened. I propose to cancel Table 1 ; the service design process has been described too detailed and is difficult to understand. Method: very detailed described, it also needs shortening; Method, line 237-238: the sentence: ‘Their upper limbs were easy to use (the bus scale was mild), which were represented by case-1 and case-2 codes respectively.’ is not clear for me. What does it mean? Method, line 296 word ‘personas’ is repeated twice Method, subsection: Cognitive Performance and Attention Examination: I doubt if the description of tools for measuring cognitive impairment of dementia not used in the study is really necessary, e.g. CAM, MoCA Method, Table 2: I propose to cancel Table 2. Results: valuable and interesting but the style is too long-winded and there are again many repetitions; it should be shortened as well. Results, line 437-444: the repetition of the text above ( line 429-436). Discussion: well written and quite concise, I like it. Conclusions: too long; I am not sure if the subsections: Potential demand, Design solution and Product verification are really necessary. I suggest removing them or shortening them significantly and adding them to the main conclusions.
Author Response
Response to Reviewer 2 Comments
We are very grateful to the reviewer2’s comments which make the results of this paper more solid. The manuscript was revised in accordance with the reviewers' comments and suggestions, followed by professional extensive English editing (English language editing by MDP, English-16242, Attachment 1). The followings are our descriptions of revision according to the reviewer2’s comments.
Patients with dementia often have multiple cognitive disorders such as inattention and memory. Attention is the key factor to complete tasks or learning outcomes. This study focuses on the horticultural treatment activities of the elderly with mild and moderate dementia in Yunlin County, Taiwan. The purpose of the study was to explore the potential needs and deficiencies of horticultural therapy activities in existing institutions, and to further develop indoor horticultural table games that can improve the attention of the elderly with mild and moderate dementia. The paper has a clear friendly structure (Introduction, Literature reviews, Method, Results, Discussion, Conclusions) and the subject is interesting and useful as the paper raises important issues regarding the activation and improvement of cognitive abilities of the elderly with mild and moderate dementia. The manuscript stands carefully developed methodology and good in- depth analysis of the results. The text is supplemented by 6 tables and 7 figures and enriched with 43 appropriate references.
However there are some issues which certainly should be improved:
Major comment
Point 1: The manuscript is too large and the style is too lengthy. It took me over 2 hours to read the entire article! This makes the text boring and the main idea and goals of the article can be easily lost. So the manuscript should be considerably shortened by at least 1/3 of the length. In addition, there are many unnecessary repetitions in the text.
Response 1: Thank you very much for your comments. This paper is rewritten and integrates the introduction with the literature discussion. At the same time, the research methods/results/discussions/conclusions are also readjusted.
Introduction: The introduction is rewritten and combined with the literature reviewed section, and the redundant words and sentences are also modified(Please referred to 1. Introduction,Line44-108). Research method: In this paper, we add a Data analysis section to make the method clearer (Please referred to Line 248-292). Discussion and Conclusion section: we also rewrite these two sections, and focus on the important conclusions of this article and delete irrelevant parts. (Please refer to Discussion and Conclusion)
Comments on individual sections:
Point 2: Introduction,
line 53: word ‘moderate’ is repeated twice Introduction,
Response: it has been modified.
line 84: word ‘mild’ is repeated twice Literature Reviews: I am not sure if this section is really necessary. Anyway it has to be at least shortened. I propose to cancel Table 1 ;
Response: Please refer to response 1, the literature review has been combined with the introduction, and table I have also been deleted.
the service design process has been described too detailed and is difficult to understand.
Response: The description of service design has been simplified, please refer to line 107-126.
Point 2: Method: very detailed described, it also needs shortening
Response 2: The description of the method has been shortened as much as possible, but data analysis sections have been added (see Line 110-292).
Method, line 237-238: the sentence: ‘Their upper limbs were easy to use (the bus scale was mild), which were represented by case-1 and case-2 codes respectively.’ is not clear for me. What does it mean?
Response: it has been modified, (please refer to Line 255-260)
Method, line 296 word ‘personas’ is repeated twice
Response:it has been modified.
Method, subsection: Cognitive Performance and Attention Examination: I doubt if the description of tools for measuring cognitive impairment of dementia not used in the study is really necessary, e.g. CAM, MoCA Method, Table 2: I propose to cancel Table 2.
Response: Only MMSE is reserved for dementia detection tool (please refer to Line184-195)
, table 2 has also been deleted (please refer to section 2.4.3)
Point 3: Results: valuable and interesting but the style is too long-winded and there are again many repetitions; it should be shortened as well.
Response: it has been modified,(Referred to Results section)
Results, line 437-444: the repetition of the text above ( line 429-436).
Response: it has been modified,,(Referred to Line330-337)
Discussion: well written and quite concise, I like it.
Point 4:Conclusions: too long; I am not sure if the subsections: Potential demand, Design solution and Product verification are really necessary. I suggest removing them or shortening them significantly and adding them to the main conclusions.
Response: it has been modified,,(Referred to Line577-622)
Attachment 1

Reviewer 3 Report
The manuscript describes the design and development of a horticultural intervention for older individuals with cognitive development. The process is described and results are presented for each stage of the process; a pre-post analysis of MMSE changes associated with a 5-week exposure to the indoor horticultural intervention is presented.
The paper is interesting and demonstrates an attempt to describe a quite complex process. However, after reading it, this reviewer feels that the manuscript has a number of important flaws:
Firstly, the rationale and implications for the study should be better developed. What is the impact of this study? Importantly, what is the relevance of this study to an international reading audience?
Secondly, and more importantly, the methodology employed in this study is unclear: the sections that attempt to describe the process followed are at times lengthy and fragmented; information on data analysis approach is poor and it is not clear exactly what was done with the participants before the actual treatment. The Method section should be better organised, more concise and there should be more clarity and transparency in terms of procedures followed. I also note that a data analysis section is currently missing.
Third, the authors claim that the developed intervention is effective in improving cognitive performance. However, this claim is not supported by the data presented: The study is underpowered by the small sample size; it does not use a control group; it employs the MMSE as a measure of attention, which brings a potential issue with construct validity; it does not provide information on the appropriateness of the statistical tests used for the data available or on the effect sizes of the changes observed.
Based on these observations, while I acknowledge that the sections related to the development of the treatment have merit, these require substantial revisions to better demonstrate a robust methodology. Also, I feel that the analyses related to the MMSE do not enrich the study, but actually limit it. A qualitative analysis of users' experiences of the treatment would have been more informative in my opinion.
I have added a list of specific comments below, which I hope will be of use to the authors to better develop this piece of work:
Abstract lines 13-16: the rationale and aim of the study should be better clarified. There is no clear link provided here between horticultural treatment and attentional issues here. If the background of the study is on cognitive disorders, how is the treatment associated to them? Abstract lines 16-19: It would be useful to provide the reader with a clearer indication of the methodology used in this study. Is this qualitative? quantitative? mixed? Multistage with different types of methods? Also, please state your sample size Abstract lines 33-36: If the changes in attention and calculation were not significant, the claim that attention can be improved (line 35) with this treatment is not supported. Please revise Introduction line 59: Please provide an example of "positive things" people with dementia could focus on. Introduction lines 79-80: I feel that the statement on attention as a priority for treatment would fit better after lines 61-62 ("Attention is a key factor.."), as it is not specifically related to horticultural treatments. Following on this, a stronger rationale for running a study on horticultural activities is needed. I seem to understand that the reason is the need for well-designed horticultural activities for attention, but this should be explicitly stated Introduction line 93: Please spell out and explain what ECare is Introduction - the aim(s) of the study is not clearly stated here, but presented in the Method section (lines 200-202). Please move to introduction Sections 2.1 and 2.2 are interesting. However, I feel that they reiterate what already said in the introduction. I would suggest to integrate section 2.1 into lines 51-62 and section 2.2 into lines 63-79, so that the information is less fragmented. If so, the section "literature reviews" could be dropped and section 2.3 could become part of the methods Line 154 - What does IDEO stand for? It is important to spell out acronyms the first time they appear in the text Section 3.1.1. lines 222-223 - For transparency, the interview schedule should be provided as a supplementary file Section 3.1.1. lines 226-227 The authors state that "after several participatory observations and interview, the target group was summed up from 14 interviews". It is unclear from this whether more than 14 participants were approached initially and then the final sample was 14, or not. Also, it is not clear what it is meant with "several". Did each participant undergo more than one observation/interview? Please clarify. Section 3.1.3. line 241 - Does "selected cases" refer to participants? Section 3.4.1. did the 20 participants involved in this phase include any of the 14 involved in the previous phase? Section 3.4.3. What type of data analysis was carried out in the different stages? What type of statistical analysis was used for the pre-post analysis? A data analysis section should be provided Section 3.4.3 - It is incorrect to say that the MMSE provides an attention score, as it is a measure of global cognitive functioning, as the authors acknowledge in section 3.4.2 Results section 4.1.3 - with what type of analysis were the various models developed? Section 4.4.1.1. the authors have used t-tests to analyse changes pre-post. However, no analyses testing normality assumptions are presented. Importantly, the effect sizes of the various t-tests should be presented, as the descriptive data indicate very small changes. Does a change in orientation from 5.14 to 6.57 have a meaning practical significance? Especially given the small sample size. Lastly, it is not clear why 2 different p-values are presented in Table 5 Section 5.3 I respectfully disagree with the authors' claim that the 5-weeks product experience "effectively" improved cognitive performance, for a number of reasons: 1) The study did not use a control group, thus, a learning effect cannot be excluded (i.e., participants learned how to complete the MMSE tasks); 2) the study has a very small sample size, thus very underpowered to talk about effectiveness; 3) reporting of effect sizes as well as checks of data assumptions would be needed to ensure that the changes observed were of practical significance and to exclude a type I error. As a minor comment, I would recommend the authors to proofread the manuscript to ensure consistency in terms of present/past/future tenses. Some statements are provided using the past, but others are described in the future.Author Response
Response to Reviewer 3 Comments
We are very grateful to the reviewer3’s comments which make the results of this paper more solid. The manuscript was revised in accordance with the reviewers' comments and suggestions, followed by professional extensive English editing (English language editing by MDP, English-16242,Attachment 2). The followings are our descriptions of revision according to the reviewer3’s comments.
The manuscript describes the design and development of a horticultural intervention for older individuals with cognitive development. The process is described and results are presented for each stage of the process; a pre-post analysis of MMSE changes associated with a 5-week exposure to the indoor horticultural intervention is presented. The paper is interesting and demonstrates an attempt to describe a quite complex process. However, after reading it, this reviewer feels that the manuscript has a number of important flaws:
Major comments:
Firstly, the rationale and implications for the study should be better developed. What is the impact of this study? Importantly, what is the relevance of this study to an international reading audience?
Secondly, and more importantly, the methodology employed in this study is unclear: the sections that attempt to describe the process followed are at times lengthy and fragmented; information on data analysis approach is poor and it is not clear exactly what was done with the participants before the actual treatment. The Method section should be better organised, more concise and there should be more clarity and transparency in terms of procedures followed. I also note that a data analysis section is currently missing.
Third, the authors claim that the developed intervention is effective in improving cognitive performance. However, this claim is not supported by the data presented: The study is underpowered by the small sample size; it does not use a control group; it employs the MMSE as a measure of attention, which brings a potential issue with construct validity; it does not provide information on the appropriateness of the statistical tests used for the data available or on the effect sizes of the changes observed.
Based on these observations, while I acknowledge that the sections related to the development of the treatment have merit, these require substantial revisions to better demonstrate a robust methodology. Also, I feel that the analyses related to the MMSE do not enrich the study, but actually limit it. A qualitative analysis of users' experiences of the treatment would have been more informative in my opinion.
Response : Thank you very much for your comments. This paper is rewritten and integrates the introduction with the literature discussion. At the same time, the research methods/results/discussions/conclusions are also readjusted.
Introduction: The introduction is rewritten and combined with the literature reviewed section, and the redundant words and sentences are also modified(Please referred to 1. Introduction,Line44-108). Research method: In this paper, we add a Data analysis section to make the method clearer (Please referred to Line 248-292). Result section: We also detect the normality for data and effect size of the data and focus the data analysis on the overall cognitive function (Please referred to Line 483-494). Discussion and Conclusion section: we also rewrite these two sections, and focus on the important conclusions of this article and delete irrelevant parts. (Please refer to Discussion and Conclusion) MMSE: This study uses MMSE, which is based on the fact that this questionnaire is one of the current evaluations of dementia medication and National Health Insurance in Taiwan, so this study uses the MMSE questionnaire to evaluate. However, more evaluation methods (i.e. CDR) of dementia should be added. Insufficient Samples: This is one of the limitations of this study, so we also analyzed the effect size to confirm its effectiveness and found that the effect is indeed low due to the insufficient number of samples (Please referred to Line 483-494)
I have added a list of specific comments below, which I hope will be of use to the authors to better develop this piece of work:
Point 1: Abstract lines 13-16: the rationale and aim of the study should be better clarified. There is no clear link provided here between horticultural treatment and attentional issues here. If the background of the study is on cognitive disorders, how is the treatment associated to them?
Point 2 : Abstract lines 16-19: It would be useful to provide the reader with a clearer indication of the methodology used in this study. Is this qualitative? quantitative? mixed? Multistage with different types of methods? Also, please state your sample size
Response 1/2: The abstract has been rewritten and each stage has been added for qualitative or quantitative analysis (please refer to Line 14-19).
Point 3:Abstract lines 33-36: If the changes in attention and calculation were not significant, the claim that attention can be improved (line 35) with this treatment is not supported. Please revise Introduction
Response 3: The abstract has been rewritten and each stage has been added for qualitative or quantitative analysis (please refer to Line 14-19)
Point 4: line 59: Please provide an example of "positive things" people with dementia could focus on.
Response 4: It has been modified(see Line 60-61).
Point 5: Introduction lines 79-80: I feel that the statement on attention as a priority for treatment would fit better after lines 61-62 ("Attention is a key factor.."), as it is not specifically related to horticultural treatments. Following on this, a stronger rationale for running a study on horticultural activities is needed. I seem to understand that the reason is the need for well-designed horticultural activities for attention, but this should be explicitly stated
Response 5: The introduction is rewritten and combined with the literature reviewed section, and the redundant words and sentences are also modified(Please referred to 1. Introduction,Line47-108).
Point 6:Introduction line 93: Please spell out and explain what ECare is Introduction - the aim(s) of the study is not clearly stated here, but presented in the Method section (lines 200-202).
Response 6: It has been modified.(Please referred to Line 102,91-95),
Point 7:Please move to introduction
Sections 2.1 and 2.2 are interesting. However, I feel that they reiterate what already said in the introduction. I would suggest to integrate section 2.1 into lines 51-62 and section 2.2 into lines 63-79, so that the information is less fragmented. If so, the section "literature reviews" could be dropped and section 2.3 could become part of the methods
Response 7: The introduction is rewritten and combined with the literature reviewed section, and the redundant words and sentences are also modified(Please referred to 1. Introduction,Line44-108)
Point 8:Line 154 - What does IDEO stand for? It is important to spell out acronyms the first time they appear in the text Section 3.1.1.
Response 8: Explained in text, (see Line 111)
Point 9:lines 222-223 - For transparency, the interview schedule should be provided as a supplementary file
Response 9: Please refer to attachment 1.
Point 10: Section 3.1.1. lines 226-227 The authors state that "after several participatory observations and interview, the target group was summed up from 14 interviews". It is unclear from this whether more than 14 participants were approached initially and then the final sample was 14, or not. Also, it is not clear what it is meant with "several". Did each participant undergo more than one observation/interview? Please clarify.
Response 10: 14 participants were interviewed only once (see line 140-149).
Point 11: Section 3.1.3. line 241 - Does "selected cases" refer to participants?
Response 11: After interviews and MMSE and Barthel index tests, two cases were observed to participate in horticultural activities (see Line 255-260).
Point 12:Section 3.4.1. did the 20 participants involved in this phase include any of the 14 involved in the previous phase?
Response 12: Participants in the product validation experiment were patients recruited from another daycare center (see Line 174-181).
Point 13: Section 3.4.3. What type of data analysis was carried out in the different stages? What type of statistical analysis was used for the pre-post analysis? A data analysis section should be provided
Response 13: we add a Data analysis section to make the method clearer (Please referred to Line 248-292).
Point 14:Section 3.4.3 - It is incorrect to say that the MMSE provides an attention score, as it is a measure of global cognitive functioning, as the authors acknowledge in section 3.4.2
Response 14: Thanks for your comments. The results focus the data analysis on the overall cognitive function (Please referred to Line 476-494).
Point 15: Results section 4.1.3 - with what type of analysis were the various models developed?
Response 15: We discuss this behavior modeling analysis in data analysis. In this study, we record and analyze the activity, environment, interactive, object and user to form these four models. (See Line 261-279).
Point 16: Section 4.4.1.1. the authors have used t-tests to analyse changes pre-post. However, no analyses testing normality assumptions are presented.
Importantly, the effect sizes of the various t-tests should be presented, as the descriptive data indicate very small changes. Does a change in orientation from 5.14 to 6.57 have a meaning practical significance? Especially given the small sample size.
Response 16: We also detect the normality for data and effect size of the data and focus the data analysis on the overall cognitive function (Please referred to Line 476-494).
Point 17: Lastly, it is not clear why 2 different p-values are presented in Table 5
Response 17: it has been modified. (Please referred to Table 5).
Point 18: Section 5.3 I respectfully disagree with the authors' claim that the 5-weeks product experience "effectively" improved cognitive performance, for a number of reasons:
Response 18: it has been modified. (Please referred to 4.3).
1) The study did not use a control group, thus, a learning effect cannot be excluded (i.e., participants learned how to complete the MMSE tasks);
Response: This is the limitation of the experimental design of this study(Please referred to 4.3).
2) the study has a very small sample size, thus very underpowered to talk about effectiveness;
Response: This is the limitation of the experimental design of this study(Please referred to 4.3).
3) reporting of effect sizes as well as checks of data assumptions would be needed to ensure that the changes observed were of practical significance and to exclude a type I error.
Response: We detect the normality for data and effect size of the data and focus the data analysis on the overall cognitive function (Please referred to Line 476-494).
As a minor comment, I would recommend the authors to proofread the manuscript to ensure consistency in terms of present/past/future tenses.
Some statements are provided using the past, but others are described in the future.
Response: English language editing by MDP, English-16242.
Attachment 1:
Semi structured interview schedule:
Interview time:
October 29 ~ November 1, 2018 Monday ~ Thursday afternoon: 1400 ~ 1630
Interview location: Chaoyang care center, Douliu City
Interviewees: 14, 3 men and 11 women.
Time of each interview: 30 minutes
Interview planning:
Open interview for 10 minutes
Interview topic:
Ask respondents to talk about the working background before retirement. Preference for gardening Interviewees talk about planting experience. Knowledge of seasonal flowers or plants in Taiwan.Break for one minute
Mini-Mental State Examination for each participants for about 20 minutes. Barthel Index scores were provided by the care center.Interview time schedule:
Time Date |
14:00~1430 |
1440~1510 |
1520~1550 |
1600~1630 |
2018/10/29 |
P1 Mr. Chen Jun-sheng |
P2 Mr. Zhuang Zhengtong |
P3 Ms. Huang Lin Azuan |
P4 Ms .Shei Chen Guihua |
2018/10/30 |
P5 Ms. Xu zhangjinzh |
P6 Ms. Zhang Yangrun |
P7 Mr. Zhou Kuncheng |
P8 Ms. Zheng Sufang |
2018/10/31 |
P9 Ms. Huang Yuxue |
P10 Ms.Lin Chunju |
P11 Ms. Chen Xi |
P12 Ms. Liu Linsu |
2018/11/1 |
P13 Ms. Huang Shijiu |
P14 Ms. Qiu Chen Boji |
|
|
Attachment 2:

Round 2
Reviewer 1 Report
Dear authors,
your paper has been significantly improved. English language and style now are quite fine. I welcome your efforts to shorten the text and to eliminate the duplications. The structure of the article is now much more refined and clear. Some of my earlier remarks (regarding MMSE for cognitive assessment, small number of samples and the lack of a control group) remain the same, however, I understand that the work is already done.
Author Response
Thanks to the valuable comments of reviewer 1, this article has made the following modifications:
- Section 2.4.2. (Experimental steps and details) has moved to supplementary materials (refer to Line 183-193).
- Product design details (Line 446-443), rewrite and simplify (refer to Line 390-400).
- The Conclusion section has been rewritten (refer to Line 513-534).

Reviewer 2 Report
Thanks to the authors for all their corrections. The paper has been significantly improved. However, in my opinion it is still too large and requires some shortening. Attached are some of my comments enclosed to the article.

Author Response
Thanks to the valuable comments of reviewer 2, this article has made the following modifications:
- Section 2.4.2(Overall Cognitive Function Examination) has been deleted, please refer to the Method section.
- Section 2.4.2. (Experimental steps and details) has moved to supplementary materials (refer to Line 183-193).
- Product design details (Line 446-443), rewrite and simplify (refer to Line 390-400).
- The Conclusion section has been rewritten (refer to Line 513-534).

Reviewer 3 Report
I thank the authors for their revisions to the paper, which I feel have improved its quality. I still think that the paper is quite long and difficult to follow, but I leave this decision with the Editor. As a minor comment, in section 2.1.1 line 149, please clarify what it is meant with "attention score"
Author Response
Thanks to the valuable comments of the reviewer 3, this article has made the following modifications:
- Section 2.4.2(Overall Cognitive Function Examination) has been deleted, please refer to the Method section.
- Section 2.4.2. (Experimental steps and details) has moved to supplementary materials (refer to Line 183-193).
- Product design details (Line 446-443), rewrite and simplify (refer to Line 390-400).
- The Conclusion section has been rewritten (refer to Line 513-534).
- It has been rewritten as the degree of degree (MMSE score), referred to as Line 147-148.